

**The influence of tillage on $N_2O$ fluxes from an intensively managed grazed**
**grassland in Scotland**
**Authors:** N.J. Cowan [a, b], P.E. Levy [a], D. Famulari [a], M. Anderson [a], J. Drewer [a], M. Carozzi [c], D.S. Reay [b], U.M.
Skiba [a]
[a] Centre for Ecology and Hydrology, Penicuik, Edinburgh, UK, EH26 0QB
[b] School of Geosciences, Kings Buildings, University of Edinburgh, Edinburgh, UK, EH9 3JG
[c] INRA, INRA-AgroParisTech, UMR 1402 EcoSys, 78850 Thiverval-Grignon, France.
Keywords: plough, greenhouse gas, nitrous oxide, gap filling
*Correspondence to*: Nicholas Cowan (nicwan11@ceh.ac.uk)
**Abstract**
Intensively managed grass production in high rainfall temperate climate zones is a globally important source of
$N_2O$. Many of these grasslands are occasionally tilled and can lead to increased $N_2O$ emissions. This was
investigated by comparing $N_2O$ fluxes from two adjacent intensively managed grazed grasslands in Scotland, one
of which was tilled. A combination of eddy covariance, high resolution dynamic chamber and static chamber
methods greatly improved the temporal and spatial coverage of $N_2O$ fluxes before and after the tillage event and
is recommended to be followed in future studies.
Total cumulative fluxes calculated for the tilled and un-tilled fields over the 175 day measurement period were
$2.45 \pm 0.27$ and $2.08 \pm 0.23$ kg $N_2O$-N ha$^{-1}$, respectively. $N_2O$ emissions from the tilled field increased significantly
for several days immediately after ploughing and remained elevated for approximately two months after the tillage
event contributing to an estimated increase in $N_2O$ fluxes of $1.08 \pm 0.14$ kg $N_2O$-N ha$^{-1}$. Cumulative fluxes
calculated over a 28 day period in August after the application of 70 kg-N ha$^{-1}$ as ammonium nitrate to both fields
were estimated at $0.42 \pm 0.15$ and $0.75 \pm 0.14$ kg $N_2O$ N ha$^{-1}$ for the tilled and un-tilled fields, respectively. The
tillage event appears to have substantially increased $N_2O$ fluxes from the tilled grassland field over a two month
period; however, this increase may have been fractionally offset by a decrease in emissions after the August
fertilisation event.



## 1      Introduction


Modern agriculture and intensive land management practices are believed to contribute over 39 % of total global
anthropogenic emissions of the greenhouse gas (GHG) nitrous oxide ($N_2O$) (IPCC, 2014). $N_2O$ is a naturally
occurring GHG released into the atmosphere by the microbial processes of nitrification and denitrification which
occur in soils and aquatic systems (Davidson et al., 2000; Seitzinger et al., 2000). Human activities which alter
environmental conditions can have a significant impact on natural microbial processes which in turn can increase
$N_2O$ emissions. Agricultural activities such as the use of nitrogen fertilisers, livestock production and land use
changes are all important sources of anthropogenic $N_2O$ from agricultural soils (Fowler et al., 2013).

There is still large uncertainty associated with the quantification of $N_2O$ emissions released from

agricultural soils on a national and global scale due to the large spatial and temporal variability of $N_2O$ fluxes
(Cowan et al., 2015; Jahangir et al., 2011; Mathieu et al., 2006). Many past experiments have measured the release
of $N_2O$ from soils after the application of nitrogen fertilisers - which are believed to be the most significant
contributor to the rise of $N_2O$ emissions since pre-industrial times (e.g. Bouwman et al., 2002; Dobbie et al.,
1999). Other causes of $N_2O$ emissions from agricultural soils, such as tillage and compaction, are less well
documented, thus preventing effective assessment of their contribution to the overall annual $N_2O$ flux from the
agricultural sector.

The use of nitrogen fertilisers (Abdalla et al., 2010; Yamulki and Jarvis, 2002), the presence of crop

residues (Baggs et al., 2003; Mutegi et al., 2010), soil compaction (Ball et al., 2008; Yamulki and Jarvis, 2002)
and the regularity and method of tillage (Sheehy et al., 2013) have all been reported to affect tillage induced $N_2O$
emissions. Changes in $N_2O$ emissions after tillage events are believed to be partly due to altering the bulk density,
water filled pore space (WFPS) and oxygen availability in soils which can lead to an increase or decrease in
nitrification and denitrification rates depending on environmental conditions (Elmi et al., 2003; Palma et al.,
1997). The addition of nitrogen to tilled soils in the form of decaying plant matter (crop residues) is a recognised
potential source of $N_2O$; however the emissions associated with specific crop residues are not well quantified.
Currently the IPCC emission inventories estimate that 1 % of all organic nitrogen applied to soils as crop residues
will be emitted in the form of $N_2O$ (IPCC, 2006).

The large number of variables which may alter microbiological processes in tilled soils can lead to a wide

variety of results between experiments carried out at different field sites under different meteorological conditions.



As a result, tillage events in agricultural fields have been reported to have very different effects on $N_2O$ production.
Some experiments have reported large increases in $N_2O$ emissions varying from 0.89 to 3.37 Kg N ha$^{-1}$ (i.e.
Chatskikh and Olesen, 2007; Merbold et al., 2014; Omonode et al., 2011; Pinto et al., 2004; Yamulki and Jarvis,
2002) dependant of fertiliser application post-tillage, whereas others have shown a zero (i.e. Boeckx et al., 2011;
Choudhary et al., 2002) or potentially negative effect of tillage (-0.88 Kg N ha$^{-1}$, Tan et al., 2009). There is no
consensus among these studies when quantifying the effect of different drivers of $N_2O$ production; however, it is
common that factors influencing the aerobicity of the soil (such as WFPS and bulk density) are cited as influential
in most tillage studies.
Improving our understanding of $N_2O$ fluxes from tillage events is important, especially in countries such
as the UK, where agriculture accounts for approximately 70 % of the total land coverage (DEFRA, 2012) and
tillage is widely practiced. Improved grasslands alone account for 25 % of the total land coverage of the UK
(Morton et al., 2011). Although permanent grasslands are tilled less often than arable land, the conversion between
arable and grazed pasture is a fairly common occurrence which requires a tillage event. Due to the large number
of tillage events which occur annually across the country it is possible that even small perturbations in fluxes
caused by these events could contribute significantly to the total national inventory of anthropogenic $N_2O$
emissions. Unlike arable fields, many of which are tilled annually, intensively managed grasslands are only tilled
occasionally (generally every 5 to 20 years depending on soil conditions and grazing intensity), either for
conversion to arable use or to improve grass sward productivity. The regularity of sward renewal depends
primarily on the condition of the grass available for grazing and desired stocking density and is entirely dependent
on the opinion and experience of farm managers in different climates. Few experiments have been carried out on
GHG emissions resulting from the tillage of grassland fields due to the infrequency of these events.  The aim of
this work was therefore (i) to use multiple $N_2O$ flux measurement methodologies to add to the understanding of
the magnitude of $N_2O$ fluxes from grasslands tilled for sward renewal, (ii) develop an improved statistical
methodology which allows for uncertainties in cumulative flux emissions to be calculated for the event, and (iii)
compare our cumulative flux emissions with those predicted using the current IPCC methodology for crop residue
emissions.






## 2 Materials and method

### 2.1 Field site

Fluxes of $N_2O$ were measured from an area of intensively managed, grazed grassland (Easter Bush, Scotland, 55°
51' 55.30"N, 3° 12' 22.17"W) before and after a tillage event on the 1st of May 2012, and were compared with
fluxes measured from an adjacent grassland which remained un-tilled (as described in Jones et al., 2011) (Figure
1). The climate is temperate maritime, with an average annual rainfall of 921 mm and average annual air
temperature of 9 °C (in the period 2001–2011). The two fields (each approximately 5.4 ha) have been managed
for intensive livestock production for at least twenty years, and since 2002 were predominately grazed by sheep.
The average stocking densities were 0.7 LSU ha⁻¹ (livestock units) and average N fertiliser application rates have
been approximately 200 kg N ha⁻¹ y⁻¹. Mainly $NH_4NO_3$ or NPK compound fertilisers were applied in three split
applications usually between March and July (Skiba et al., 2013).

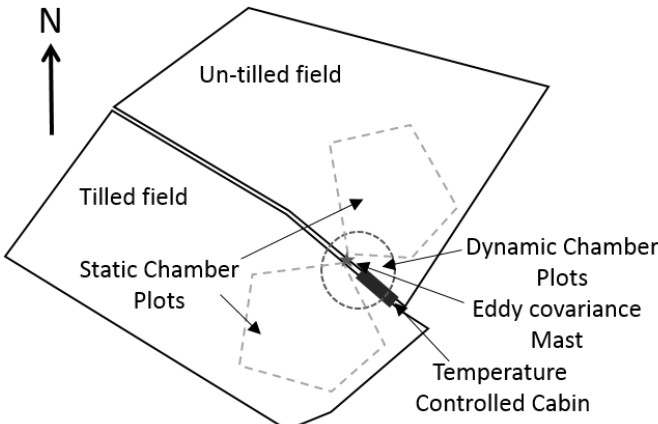

**Figure 1** $N_2O$ fluxes were measured from two adjacent grassland fields at the Easter Bush Farm (Penicuik,
Scotland). The north field remained un-tilled, while the south field was ploughed on the 1st of May 2012. An eddy
covariance mast was setup next to a permanent cabin positioned between the fields. Dynamic chamber
measurements were made within a 30 m radius of the cabin. Static chambers were spread out within the fetch of
the eddy covariance mast.





The soils are clay loams with a sand/silt/clay texture of 28/20/52 and 24/19/57 for the top 30 cm in the
un-tilled and tilled fields, respectively with a pH of 5.1 (in $H_2O$). They are classed as an imperfectly drained
Macmerry soil of the Rowanhill association (eutric cambisol, FAO classification). A drainage system had been
installed about 50 years ago, but is no longer functioning well, resulting in frequent occurrence of surface water
during rainy periods. The fields had not been tilled for at least twenty years, and the farmer had reported reduced
fertility and productivity. This together with the poor drainage led to the decision to till both fields.
In the first stage, only one field (also called the South Field in Jones et al., 2011) was tilled (Table 1). In
preparation, glycophosphate (1.5 l ha$^{-1}$) was applied to kill the grass three days prior ploughing on the 27$^{th}$ of
April. The addition of glycophosphate to the soil is considered standard practice when grassland fields are
ploughed in the area and the effect that it may have had on microbiological processes or the decay of the grass
materials ploughed into the soil are considered to be part of the tillage event as a whole in this study. The field
was ploughed to a depth of 30 cm on the 1$^{st}$ of May 2012. Two days after ploughing the field was harrowed, then
rolled and sown with ryegrass (*Lolium perenne L.*) three days after ploughing. The un-tilled field (also called the
North field in (Jones et al., 2011)) was managed as usual and grazed by sheep (approximately 30 sheep ha$^{-1}$).
Fertilisation events continued as normal on the un-tilled field which received two ammonium nitrate (Nitram)
fertiliser applications of 70 kg-N ha$^{-1}$, one on the 28$^{th}$ of May and the second on the 9$^{th}$ of August. The tilled field
only received a 70 kg-N ha$^{-1}$ Nitram application on the 9$^{th}$ of August approximately four months after the tillage
event.
**Table 1** Field management events for both the tilled and un-tilled fields in 2012.

| Date | Tilled Field (South) | Un-Tilled Field (North) |
|---|---|---|
| 27$^{th}$ April 2012 | Glycophosphate application (1.5 l ha$^{-1}$) | |
| 1$^{st}$ May 2012 | Ploughing at 30 cm depth | |
| 3$^{rd}$ May 2012 | Harrowing, seeding & rolling | |
| 28$^{th}$ May 2012 | | 70 kg-N ha$^{-1}$ Nitram application |
| 9$^{th}$ August | 70 kg-N ha$^{-1}$ Nitram application | 70 kg-N ha$^{-1}$ Nitram application |


Biomass samples were collected from the fields before tillage. Twenty soil cores (12 cm deep and 5.8
cm diameter) were extracted from the fields. All above ground biomass was removed from the soil and dried in





an oven at 80℃ until constant weight. Once dry the above ground biomass was weighed. The remaining soil cores
were broken up by hand and dried at 100℃ until constant weight. After drying, the root materials were separated
from the soils by hand and weighed. Sub samples of the dried plant materials were prepared for combustion
elemental analysis of total carbon and nitrogen contents (vario EL cube, Elemaentar, Hanau, Germany).

### 129   2.2      Flux Measurements

$N_2O$ fluxes were measured from both tilled and un-tilled fields over a seven month period using three measurement
methodologies; eddy covariance, static chamber and dynamic chamber techniques. Eddy covariance was the
primary measurement methodology used; however, due to unpredictable changes in wind direction at the site it
was necessary to deploy chamber methodology to ensure that both fields were measured periodically during the
experiment. The dynamic chamber measurements was used as a cost effective way to provide many (>30) high
resolution $N_2O$ fluxes on the days immediately after tillage without the need for time consuming GC lab analysis
required by static chambers.

An eddy covariance system was installed on the 27[th] of March on the field boundary (See Figure 1).  An

ultra-sonic anemometer (WindMaster Pro 3-axis, Gill, Lymington, UK) mounted at 2.2 m was used to measure
fluctuations in 3-D wind components at a frequency of 10 Hz. Mixing ratios of $N_2O$, $H_2O$ and $CO_2$ were measured
at 10 Hz by a quantum cascade laser (QCL) gas analyser (CW-QC-TILDAS-76-CS, Aerodyne Research Inc.,
Billerica, MA, USA), housed in a temperature controlled cabin. The inlet line to the QCL was a 13.5 m length of
Dekabon tubing (0.25 inch outer diameter), with a flow rate of approximately 13 l min[-1].  Fluxes were calculated
at 30 min intervals using the EddyPro software (Version 5.2.1) (Li-Cor, Lincoln, NE, U.S.A.), based on the
covariance between the $N_2O$ concentration ($\chi$) and vertical wind speed (w):

$$F_{\chi=\overline{\chi'w'}}$$                                  (Eq. 1)

In the processing, we applied double coordinate rotation (vertical and crosswind), spike removal, block

averaging, and time lag removal by covariance maximisation. Correction for the frequency response of the system,
both high and low-frequency losses, were made using the method of Moncrieff et al., (1997). Corrections for
density fluctuations were applied on a half-hourly basis using the method of (Burba et al., 2012).  The quality
control scheme of Mauder & Foken, (2004) was used to remove poor quality flux measurements (their category
2).  Data were also rejected on the basis of high instrumental noise, and friction velocity ($u*$) values less than 0.05



m s$^{-1}$. Fluxes measured with a mean wind direction between 150 and 300 degrees were classed as from the tilled
field; those measured at greater than 320 and less than 130 degrees were classed as from the un-tilled field. The
remaining data were disregarded due to obstruction of the wind by the cabin and fence line.  Standard
meteorological variables (rainfall, air temperature and soil temperature) were recorded by a nearby weather
station.

N$_2$O was also measured from both fields using static chamber and dynamic chamber techniques. The

static chambers consisted of a cylindrical polyvinyl chloride (PVC) plastic pipe of 38 cm inner diameter (ID) and
22 cm height. These chambers were inserted 5 cm into the soil, giving a headspace of approximately 20.4 l.
Chambers were closed for 40 mins, during which time three 100-ml gas samples were collected via a syringe and
a three-way tap fitted to the lid, at t = 0, 20 and 40 mins. After each measurement, chamber height was measured
at five points to estimate the chamber volume.  Gas samples were stored in 20 ml glass vials which were flushed
with 100 ml of air in the syringe using a double needle. Samples were analysed using a Hewlett Packard 5890
series II gas chromatograph (Agilent Technologies, Stockport, fitted with an electron capture detector) (Skiba et
al., 2013).

Ten static chambers were positioned in each of the fields, within the estimated flux footprint of the eddy

covariance system (10 to 200 m from the mast). Static chambers in the un-tilled field were kept in the same
position for the duration of the experiment. Chambers in the tilled field were occasionally moved to allow access
to farm vehicles during the different stages of the tillage operation. Chamber measurements were carried out
between 9:00 and 15:00 on the measurement dates. Fluxes were calculated as:
$$F = \frac{d\text{C}}{d\text{t}} \cdot \frac{\rho V}{A}$$
(Eq. 2)

where $F$ is the gas flux from the soil (nmol m$^{-2}$ s$^{-1}$), $dC/dt$ is the rate of change in concentration with time in nmol
mol$^{-1}$ s$^{-1}$ estimated by linear regression, $\rho$ is the density of air in mol m$^{-3}$, $V$ is the volume of the chamber in m$^3$
and $A$ is the ground area enclosed by the chamber in m$^2$.

Fluxes were also measured using the QCL in a closed, dynamic chamber system (Cowan et al., 2014a).

A chamber (39 cm ID, 22 cm high) was placed onto a stainless steel collar inserted several cm into the soil (on
average 5 cm) prior to measurement. Two 30 m lengths of 3/8 inch ID Tygon® tubing connected the chamber to
the inlet of the QCL and the outlet of a vacuum pump (SH-110, Varian Inc, CA, USA) to form a closed system.



This allowed a 30-m possible radius from the instrument cabin in which the chamber could be placed (Figure 1).
A flow rate of approximately 6 to 7 L min$^{-1}$ was used, with a lag time of approximately 22 seconds between the
chamber and analyser. Fluxes of $N_2O$ were calculated with 1-Hz data over three minutes, using both linear and
non-linear asymptotic regression methods (Levy et al., 2011; Pedersen et al., 2010). Using a mixture of goodness-
of-fit statistics and visual inspection, the regression method that provided the best fit for the time series of mixing
ratios of $N_2O$ was chosen for each individual measurement. The detection limit of individual fluxes calculated by
this method was approximately 0.04 nmol m$^{-2}$ s$^{-1}$ compared to 0. 4 nmol m$^{-2}$ s$^{-1}$ when using the static chambers
(Cowan et al., 2014a, 2014b).

In the first few days after the tillage event, the wind direction was north-easterly, meaning that the eddy

covariance system could not record fluxes from the tilled field (to the south-west). The dynamic chamber
measurements were primarily used to fill this gap in the eddy covariance time series with high precision chamber
measurements.

**2.3    Gap filling**
Because the eddy covariance system was placed on the field boundary, observations could only be made on a
single field at any given time. Furthermore, some data were missing because of instrument failure and some had
to be rejected according to the quality control criteria used. In order to estimate cumulative fluxes from both
fields, temporal interpolation of the missing data points was required. However, in the absence of a well-validated
process-based model for $N_2O$ fluxes on which to base predictions, it is not obvious how this is best achieved. The
most common approach is to linearly interpolate in time between flux measurements. In this study, a general
additive model (GAM) was used as an alternative approach, which accounted for temporal patterns at a range of
time scales and nonlinear responses to environmental variables, implemented using the *mgcv* package in the R
software (Wood, 2006).

We fitted the GAM with the same model terms to the separate data sets from the tilled and un-tilled

fields. The terms included were air temperature, soil temperature, precipitation, and time. Additional terms for
temperature and precipitation aggregated over longer intervals (1, 6, 12, 24 and 48 hours preceding the flux
measurement) were examined and included where they improved the fit. The GAM allows for non-linearity by



fitting a smooth response with cubic splines. The degree of smoothing is optimised by the algorithm, but was also
adjusted subjectively, such that the model was not over-fitting to noise in the data. Observations from eddy
covariance and the two chamber methods were given equal weighting.

Predictions from the GAM were used to fill gaps when observations were not available. Uncertainty in

predictions was estimated by simulating 2000 replicate time series from the GAM, given the uncertainty in the
fitted parameters, to estimate the posterior distribution. The quantiles of this posterior distribution provided the
95 % credibility interval at each predicted time step. To calculate cumulative fluxes, observed fluxes were used
with their associated uncertainties when available; otherwise the GAM predictions were used.

**3      Results**
**3.1      Meteorological data**
A total of 1191 mm of rain was recorded in 2012, higher than the average annual rainfall of 921 mm (2001 to
2011) for the Easter Bush area (Figure 2a). Historically, the wind direction at the field site is predominantly south-
westerly (85 %). However, during the measurement campaign, the wind direction was split more evenly between
the tilled and un-tilled fields (Figure 3). This allowed a better basis for comparison of $N_2O$ fluxes from the two
fields, although data coverage for each field was low, 27 % and 25 % for tilled and un-tilled respectively.



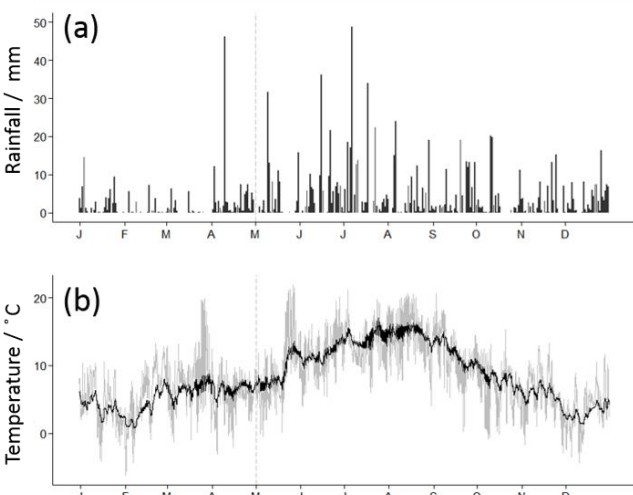


**Figure 2** (a) Accumulated daily rainfall at the Easter Bush Field site during the year 2012. (b) Air temperature at
height 3 m (grey) and soil temperature (black) recorded at the Easter Bush field site during the year 2012. Tillage
occurred on the 1st of May 2012 (grey dashed vertical line).

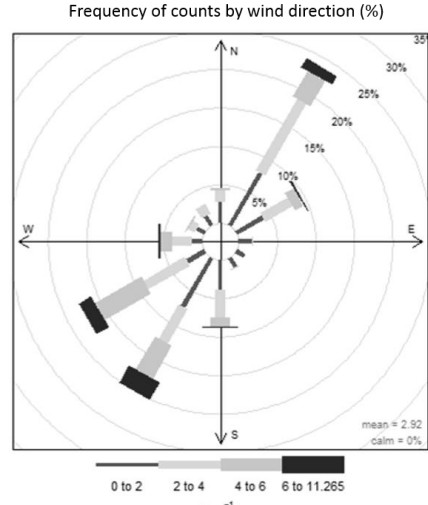


**Figure 3** Wind rose plot for the Easter Bush field site during eddy covariance measurements (March – October

2012).



### 3.2 Comparison of N$_2$O fluxes measured from the un-tilled and tilled fields

Before the tillage event, N$_2$O fluxes were similar in the tilled and untilled fields. In both cases, around 90 % of fluxes were below 0.5 nmol m$^{-2}$ s$^{-1}$ (Figure 4). All three fertilisation events (the two fertiliser events in the untilled filed and single fertiliser event in the tilled field) were characterised by an emission peak of 5-10 nmol m$^{-2}$ s$^{-1}$ lasting a few days, which declined over the following days and weeks, often with considerable variability and some apparent secondary peaks (Figure 5). Fluxes had returned to background levels (<0.5 nmol m$^{-2}$ s$^{-1}$) within 28 days of each of the fertilisation events. Fluxes measured by all methods agreed reasonably well in magnitude, and there is no strong evidence for a systematic bias, given the differences in the spatial and temporal sampling (for a more specific insight see e.g. Cowan et al., 2014a).

The tillage event also produced an increase in emissions, and although the peak was less clearly defined, the effect was more prolonged. Fluxes generally ranged from ~0 to 1.0 nmol m$^{-2}$ s$^{-1}$ in the days before tillage and ~0 to 8.8 nmol m$^{-2}$ s$^{-1}$ in the week immediately after tillage (Figure 4b). Fluxes from the tilled field from mid to late May were approximately 1 nmol m$^{-2}$ s$^{-1}$ higher than from the untilled field (before the latter was fertilised). There followed an apparent increase in N$_2$O fluxes lasting approximately four weeks from the tilled field from late May to late June, peaking in the first week in June (Figure 5c). Unfortunately, data coverage was rather low during this period due to changes in wind direction and a one week period in which the QCL was unavailable. Because the tilled field had not been fertilised since the previous year, we infer that the increased fluxes were a result of the tillage event. Fluxes in the tilled field returned to pre-tillage magnitude during July. By July, a new sward of grass had grown in the tilled field, but sheep were not re-introduced into the field until late August.

The GAM method was used to gap fill flux data to calculate cumulative fluxes (Figure 6). Cumulative N$_2$O fluxes calculated for the tilled and un-tilled fields for the 175 days from 27th of March to the 18th of October were 2.45 ± 0.27 and 2.08 ± 0.23 kg N$_2$O-N ha$^{-1}$, respectively (Figure 7).





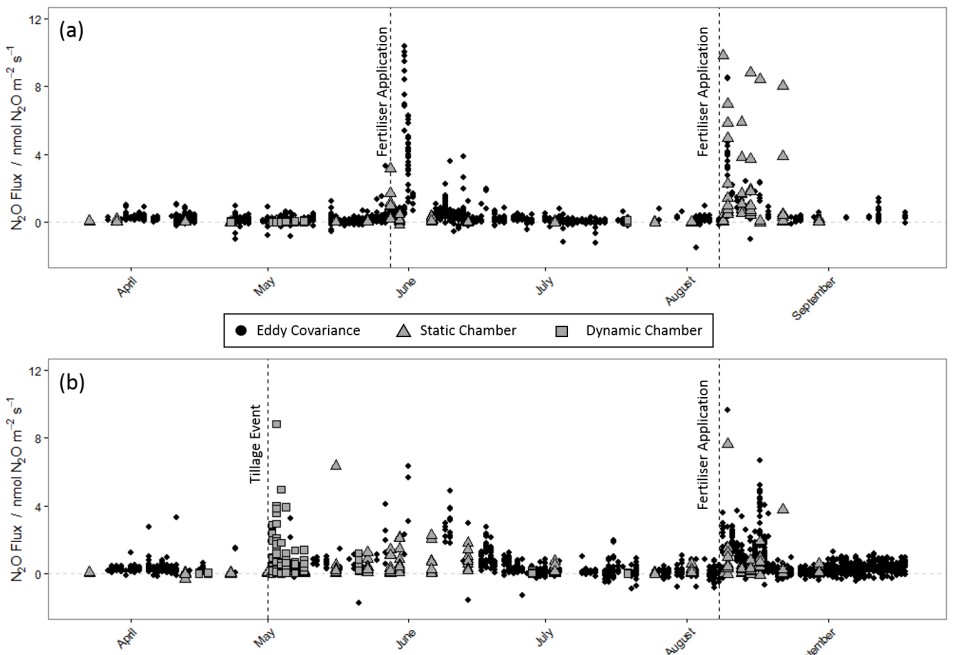

**Figure 4** Fluxes of N$_2$O from the (a) un-tilled and (b) tilled fields measured at the Easter Bush field site in 2012.

Fertiliser was applied to the un-tilled field on the 28$^{th}$ of May and to both fields on 9$^{th}$ August. Tillage began on

1$^{st}$ May. The Y-axis is limited to 12 nmol m$^{-2}$ s$^{-1}$ for better comparison between the fields. Only four static chamber

measurements in the north field recorded fluxes above 12 nmol m$^{-2}$ s$^{-1}$ in the first few days after the August

fertilisation.






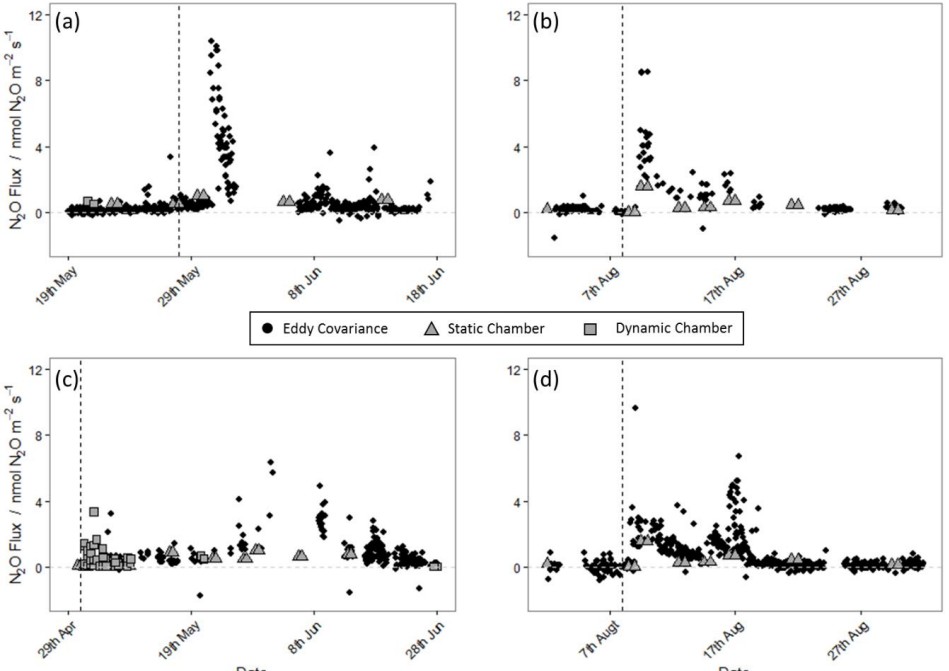


**Figure 5** Elevated fluxes of $N_2O$ were measured from the un-tilled field after (a) a 70 kg-N ha$^{-1}$ application of
Nitram on the 28$^{th}$ of May and (b) a second 70 kg-N ha$^{-1}$ application of Nitram on the 9$^{th}$ of August. Elevated
fluxes of $N_2O$ were measured from the tilled field (c) immediately after the tillage event on the 1$^{st}$ of May which
remained above pre-tillage magnitude until the end of June. Elevated fluxes of $N_2O$ were also observed from the
tilled field (d) after a 70 kg-N ha$^{-1}$ application of Nitram on the 9$^{th}$ of August.





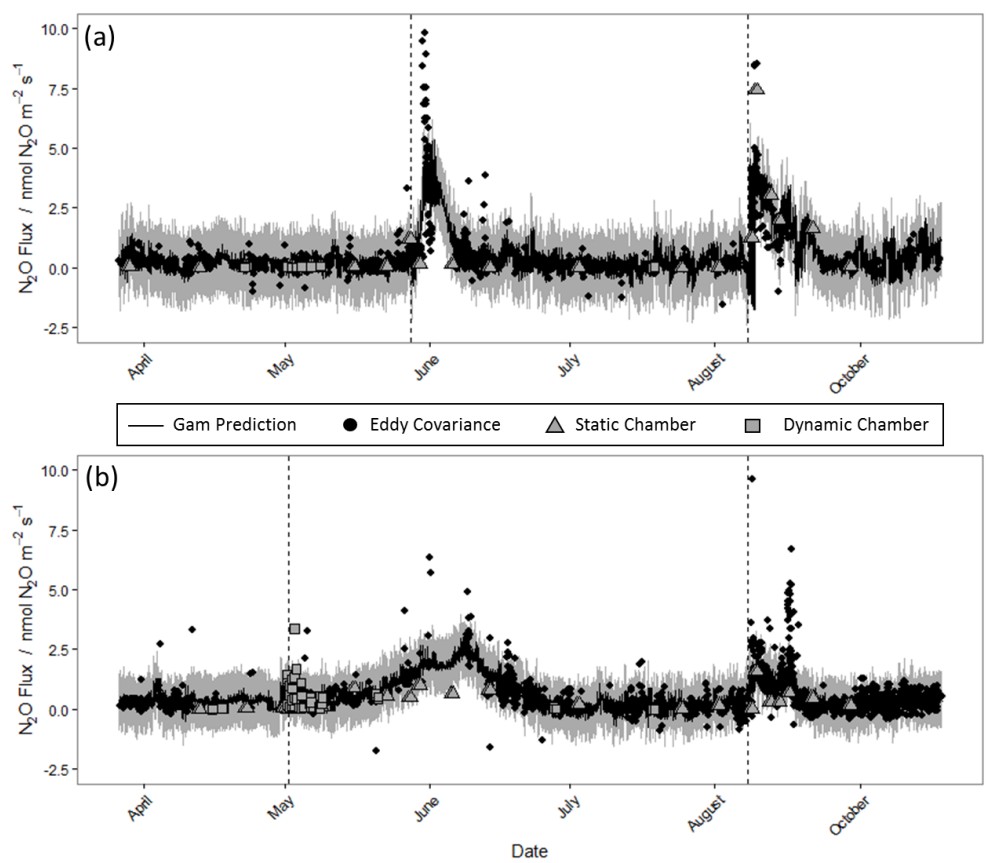


**Figure 6** The GAM method provides an estimated N$_2$O flux which can be used to gap fill measurements from

both the (a) un-tilled and (b) tilled fields at 30 min intervals. The 95 % confidence interval in the estimated flux

reported by the GAM is included (grey). Tillage and fertiliser dates are indicated (vertical lines)






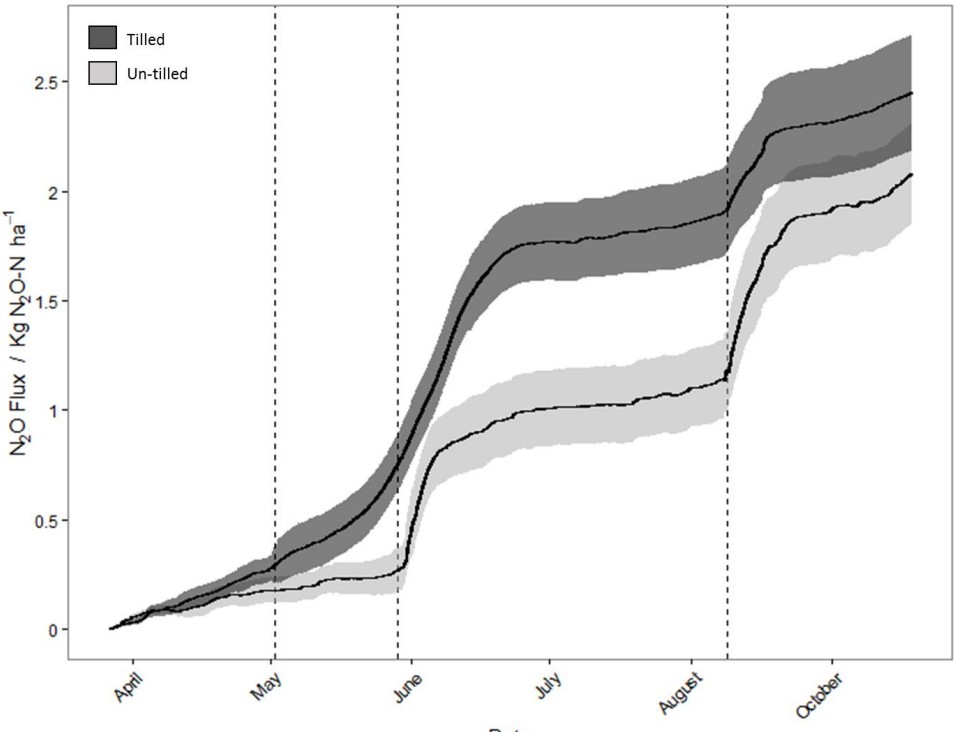


**Figure 7** Cumulative flux is calculated for the tilled (dark) and un-tilled fields (light) using the gap filled flux

data. The propagated 95 % confidence intervals estimated using the sum of least squares method (grey areas).

Fertiliser was applied to the untilled field on the 28th of May and to both fields on the 9th of August and tillage

occurred on the 1st of May (black dashed vertical lines).

**3.3**     **Biomass measurements**
Biomass measurements made from the tilled field before tillage estimated that there was an average of 369 ± 310
g m$^{-2}$ of grass materials (dried weight) growing across the field with a root to shoot ratio of ~1.5. The elemental
analysis of the dry grass materials measured an average total carbon content of 45.7 % and nitrogen content of 2.5
%. Based on these measurements it is estimated that the tillage event added a total of 93.6 kg ha$^{-1}$ of nitrogen to
the field in the form of crop residues.





**4        Discussion**
**4.1        The influence of tillage on $N_2O$ fluxes**
The comparison of pre-tillage and post-tillage fluxes from the tilled field suggests that the tillage event was
directly responsible for an immediate increase in $N_2O$ fluxes (Figure 4b & 5c). $N_2O$ fluxes significantly larger
than those measured pre-tillage were observed from the tilled field over two separate periods during which no
changes in $N_2O$ fluxes were observed in the adjacent un-tilled field. The initial increase in $N_2O$ flux from the tilled
field occurs directly after the disturbance of the soil caused by ploughing and harrowing.  In the two month period
in which fluxes from the tilled field were elevated, a total of $1.47 \pm 0.16$ kg $N_2O$-N ha$^{-1}$ was estimated to have
been released. Assuming fluxes in the tilled field had remained at approximately pre-tillage magnitude (~0.27
nmol m$^{-2}$ s$^{-1}$) had the tillage event not taken place, it can be concluded that the tillage event contributed to an
additional $1.08 \pm 0.14$ kg $N_2O$-N ha$^{-1}$ emitted from the field over a two month period.

Increases in $N_2O$ flux lasting up to two months after grassland tillage events have been observed before

in other studies using both static chamber and eddy covariance measurements (Chatskikh and Olesen, 2007;
Merbold et al., 2014). Reported fluxes can be relatively high over a sustained period of time (several days or
weeks) and similar in magnitude to those recorded after fertilisation events. The mechanisms driving these large
sustained fluxes are believed to be partly due to the mineralisation of organic materials in the soils (decaying grass
materials from the previous sward in tilled grasslands) (Baggs et al., 2003; Hellebrand, 1998; Pimentel et al.,
2015). The large quantities of decaying organic matter ploughed into the soils would have provided a gradual
release of carbon and nitrogen into the soils, which provide substrate for the microbial processes of nitrification
and denitrification (Pimentel et al., 2015; Seastedt et al., 1992). According to IPCC estimates, 1 % of N added to
soils in the form of crop residues can be expected to be released as $N_2O$ (IPCC, 2006). Based on our pre-tillage
biomass measurements (93.6 kg N ha$^{-1}$) if these estimates were true we would expect to see $N_2O$ fluxes of
approximately 0.94 kg $N_2O$-N ha$^{-1}$ from the field. This estimated value is within the range of uncertainty of our
calculated cumulative fluxes in this study ($1.08 \pm 0.14$ kg $N_2O$-N ha$^{-1}$). High emissions from crop residues tilled
into arable crops have been recorded in similar wet soils with high clay content (Ball, 1999) which may indicate
a similar process is occurring under these conditions at other field sites in the area.

Large $N_2O$ fluxes (> 0.5 nmol m$^{-2}$ s$^{-1}$) are observed from both fields after fertilisation events. Elevated

fluxes recorded from the fields after fertilisation typically last three to four weeks with an occasional large spike



lasting 24 to 48 hours before returning to pre-fertilisation levels. This month long period in which the majority of
large fluxes occur after fertilisation is also generally observed by other similar studies from the local area (Skiba
et al., 2013; Smith et al., 2012). Assuming the majority of $N_2O$ emitted after a fertilisation event occurs within a
28 day period after the fertiliser application, the 28 day cumulative flux emissions associated with the fertilisation
events on the 28th of May and 9th of August on the un-tilled field were $0.72 \pm 0.14$ and $0.75 \pm 0.14$ kg $N_2O$-N ha⁻
¹, respectively. The 28 day cumulative flux emissions associated with the fertilisation event on the tilled field was
$0.42 \pm 0.15$ kg $N_2O$-N ha⁻¹. As each fertilisation event consisted of the same application rate of 70 kg N ha⁻¹ of
Nitram pellets, these cumulative fluxes account for a 1.03, 1.07 and 0.60 % of the total applied nitrogen in each
case. Assuming the 28 day periods account well for the emission factors of the fertiliser events, these results agree
well with the generic 1 % value reported by the IPCC for N fertiliser events (IPCC, 2014).

The reason for the large difference in fluxes associated with the fertiliser events between the two fields

in this study is unknown. As the weather was the same for both fields it is assumed that temperature and rainfall
was not a factor. After sward renewal on the tilled field, the grass grew back very thick and appeared much
healthier than the more established grass in the un-tilled field. One theory is that the healthier, more productive
grass in the tilled field consumed more of the available fertiliser than in the un-tilled field, leaving less N available
for $N_2O$ producing microbial processes. Another possible explanation is that physical changes to the soil since
tillage (such as bulk density, WFPS and soil organic carbon content) altered the microbial processes in a way that
reduced $N_2O$ emissions from the fertiliser event (Ball et al., 2008; Choudhary et al., 2002; Davidson et al., 2000;
Turner et al., 2008). From the limited soil data available, it is not possible to say for certain why fluxes from the
tilled field in August were lower than those recorded in the adjacent field. As the driving forces of $N_2O$ emissions
from agricultural soils are not understood well in general, it complicates our ability to comprehend what is
happening at the microbial level.
**4.2**      **Gap filling of $N_2O$ fluxes**
Gap-filling $N_2O$ flux measurements is difficult due to the lack of reliable process-based models on which to base
predictions. $N_2O$ fluxes are believed to be driven primarily by the availability of nitrogen compounds in the soils
(ammonium and nitrate) (Davidson et al., 2000) as well as physical properties of the soil such as WFPS, aerobic
extent, soil type, temperature and compaction (Ball et al., 2008; Choudhary et al., 2002; Davidson et al., 2000;
Turner et al., 2008). The collection of these data on a temporal/spatial scale which would allow these models to
be applied is not often logistically possible or affordable. The GAM method used in this study incorporates



readily-available meteorological data with the temporal pattern in the data, to provide an empirical but practical
means of temporal interpolation, which makes use of more information than simple linear interpolation. Although
the GAM method has proved useful, we would also emphasise the dangers of extrapolating to conditions beyond
those to which the model was fitted. For example, as we have not measured fluxes during the cold months in
winter, the GAM is unable to reliably predict fluxes in temperatures lower than those measured during the study.
A two week gap in measurements in early June when the QCL instrument was out of commission has left a large
gap in data, which makes a considerable contribution to the total uncertainty in the cumulative flux following the
tillage event.

In this study spatial variability was not explicitly accounted for in the cumulative flux uncertainty and

this remains a potentially large error. Eddy covariance is able to integrate over a large area of the field (several
100 m$^2$) but these measurements are still subject to an element of spatial variability which is unaccounted for. Any
study which plans to report cumulative flux estimates should consider how to minimise the uncertainties which
arise when interpolating and/or extrapolating measurements to larger temporal and spatial scales (e.g. from
occasional chamber measurements to annual field-scale emissions). Further studies may require more complex
statistical analysis of uncertainties using methods such as Bayesian statistics to improve uncertainty estimates in
methodology.

**5     Conclusion**
Total cumulative fluxes calculated for the tilled and un-tilled fields over a 175 day period were 2.45 ± 0.27 and
2.08 ± 0.23 kg $N_2$O-N ha$^{-1}$, respectively. $N_2$O emissions after tillage were relatively large and sustained, similar
in magnitude to a nitrogen fertilisation event. The tillage event is estimated to be responsible for emissions of 1.08
± 0.14 kg $N_2$O-N ha$^{-1}$ over a two month period after tillage. Further differences in $N_2$O fluxes were observed
between the tilled and un-tilled fields after a subsequent nitrogen fertilisation in August 2012. Cumulative fluxes
of $N_2$O estimated for a four week period after fertilisation for the tilled and un-tilled fields were 0.42 ± 0.15 and
0.75 ± 0.14 kg $N_2$O-N ha$^{-1}$, respectively. It is uncertain whether the tillage event or other factors were the reason
for lower emissions from the tilled field after the fertilisation event. The results reported in this study agrees with
several other studies that nitrogen added to soils in the form of crop residues may contribute to high emissions of
$N_2$O similar to those expected from fertiliser events. This observation highlights a potentially large un-quantified
and poorly understood source of anthropogenic $N_2$O emissions at a global scale. The study also highlights the




need for more detailed investigation of the microbiological processes in soils to describe processes driving
emissions from tilled soils and crop residues which also remain poorly understood.
**6      Acknowledgements**
We thank farm manager Wim Bosma, for the Easter Bush field site, who provided us with the opportunity to carry
out this experiment. We thank DEFRA and the UK Devolved Administrations for financial support through the
UK GHG Platform project AC0116 (The InveN2Ory project). We also thank the INGOS EU funded Integrating
Activity for support of the field infrastructure.

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
