# Peer review of "The influence of tillage on N2O fluxes from an intensively managed grazed"

_Biogeosciences, 2015_

## Referee Comment (RC1) · L Merbold (Referee) · 25 Feb 2016

General remarks: The authors present N2O flux data from a permanent managed grassland in Scotland. Thereby a special focus was given to the effects of tillage on fluxes including pre- and post ploughing effects as well as fertilization effects. A commonly used approach to measure N2O fluxes are static chambers, which were applied in this study. Additionally N2O flux measurements were carried out using dynamic chambers as well as eddy covariance measurements. The study resembles a modified "paired design" study where one side of the grassland was tilled and the other was not. Both areas were covered by chamber and EC measurements leading to a comprehensive dataset with few gaps. A task which is not easily achieved during observation studies. The authors describe and discuss about potential reasons for enhanced N2O emissions following tillage and fertilization and provide a solid approach to fill data gaps

when having sufficient training data. Overall the manuscript is written in a concise and understandable way and the figures are neatly prepared. I have few major comments and provide minor/technical corrections below.

Major Comments: (1) There are yet not too many N2O fluxdatasets from grassland available and even less from tillage. I am wondering why the authors make such a short story out of this valuable dataset. This study is based on three different techniques and the actual discussion on the capability of these techniques to capture the same results differ. Even though the temporal pattern seems to be captured well, there seems also a stable offset (Figure 5) between EC and chamber measurements. This should be further elaborated. (2) When using an approach as in this study, I wonder how the authors assure that the flux from the specific 30min of EC observations originated for consecutive 30min from the area of observation? Did the authors include specific tests to proof that the approach is suitable? (3) A gapfilling approach for N2O data is presented which is suitable to be applied at many other N2O EC sites. Therefore I suggest to further elaborate on this. (4) The authors provide cumulative N2O fluxes for 175days of gapfilled N2O data. However it remains unclear how these values shall be set in context with other studies where growing season budgets or annual budgets are presented. Is there any additional data or knowledge that could lead to annual numbers?

These 4 points would make the paper a novel approach to understand N2O emission in ears of tillage.

Specific comments and suggestion for improvement: L58ff: the number provided are focusing on a specific time period but this is not indicated, please clarify. Also write kg instead of Kg L62: replace ";" by "." L63: aeration instead of aerobicity L124: aboveground L131: I suggest to add manual or automatic since this is unclear L134: is that an experiment or rather a survey? see also Eugster and Merbold et al. 2015, SOIL 1 for clarification L134ff: ">30" - dynamic doesn't mean per se automatic, but it remains unclear how you achieved this, so the needed information should be provided

here already. L152: did you check whether the wind was blowing from the specific area for the specific half-hour? L155: Specs should be given for rainfall, temperature etc measurements: where, how etc? or refer to another paper (Drewer et al. Plant and Soil for instance L176: ID? L177: "prior to measurement" - how long before? L189: how can you assure that this method is suitable to fill the gap? Details, validation is needed. L197: I disagree – Spacsys and PaSIM don't perform too badly – So I suggest to give a reference for your statement L199ff: Can extend this section so that others can choose a similar approach? L202: same model terms compared to what? L232: Where is this shown, similar on which time interval? stats? L234: detection limit of your system? L236: what is your background flux and how was it determined? L237ff: how comparable are these cumulative values to other studies? is that the main growing season, how will the fluxes be during the remaining season? L249: were sheep in the untilled field? L251: how comparable are these cumulative values to other studies? is that the main growing season, how will the fluxes be during the remaining season? L275: "were estimated" or "estimated by least square method?" L279: what do you try to state with the estimated? L283: Was all plant material incorporated or was the field grazed beforehand? when were the biomass samples taken? L287-L295: This is basically a repetition of already presented results. L311: What is your definition of large in comparison to 0.5nmol m-2 s-1 L321: What about N deposition and mineralisation the soil? L325: any proof or measurement here? Pictures, CO2 fluxes or likewise? L340: include Butterbach-Bahl et al. 2013 – review here L352: please explain? references? L363ff: no need to restate results - how about setting your results into perspective?

Figure 1: Was there any bias to the dynamic chamber if these are soo close to the cabin? Figure 2: height/depth of measurements? Figure 3: so everything adds up to 100%, correct? NNE high winds only in 2% of the data? correct? Figure 4: why are the EC fluxes all in one line after the fertilization event, are these 30min data or which averaging interval did you use? triangles and squares in grey are difficult to distinguish. What about adding an axis break including the higher fluxes? how much higher than

12? Figure 5: I suggest in all figures to change the unit indication to "( )" instead of "/"
Figure 6: This looks really good but I can not see the GAM line in much detail

I hope the comments are helpful and look forward to seeing this paper being published soon. An on-the-fly commented pdf file is attached and if you have question please do not hesitate to contact me.

with kind regards

Lutz Merbold

---

## Referee Comment (RC2) · E. Diaz Pines (Referee) · 2 Mar 2016

I have read the manuscript from Cowan and colleagues on the effects of tillage on the N2O fluxes of a managed grassland in Scotland. The work applies a combination of EC and dynamic and static chamber measurements over a growing season on a tilled and an untilled field. The tilled plot showed higher N2O emissions after ploughing, but the tilled plot was less responsive to fertilization than the untilled one (0.4 vs 0.75 kg N ha-1), partially offsetting the overall differences between tilled and untilled plots. The topic of the work is highly suitable for the intended SI (GHG hotspots): tillage of grasslands and its effect on N2O fluxes is not well addressed in the literature (presumably because ploughing in grasslands is not done yearly and relies on the managers' opinions, as the authors point out), despite the fact that it can constitute a significant source of N2O. The manuscript is of high quality, well elaborated and structured, although there is still

some room for improvement prior to final publication.

The editor specifically asked my opinion about 1) the potential of the work for scaling up the measurements and 2) the suitability of the temporal coverage. Regarding 1): I do not recommend extrapolating results of a single field without solid arguments to do it, but this work can surely contribute to the growing body of information on the issue; from this point of view, a paper is solid even if no extrapolation can be implemented (as it is in this case study). Regarding 2) (temporal coverage): the length of the measurement campaign is enough to obtain sound conclusions on the effects of tillage and fertilization events (all applications throughout the year were monitored) so that both plots can be consistently compared; but authors should also try to reflect how they could come up with annual fluxes and what would be the uncertainty there (e.g. are there soil freeze-thaw events?); in other words, authors should mention what are we (likely) missing in the 190 days which were not measured. Spatial and temporal variability is somehow addressed in the discussion (L350-357), but could be more elaborated, and some information on spatial variability (as assessed by chamber measurements) should be included.

I recommend making a stronger point in the discussion addressing the implications of tillage for accounting of N2O fluxes in the context of grassland management, including the fundamental fact that tillage (responsible for an additional release of 1.08 kg N ha-1) "substitutes" one fertilization event (responsible for 0.72 kg N ha-1, L317). For instance, confidence intervals of N2O emissions from tilled and untilled plots overlap by the end of the season (Figure 7), so I presume differences in cumulative fluxes were not statistically significant; that should be also addressed clearly in the work.

My second major point is oriented to methods. I missed some necessary explanations on 1) whether footprint analyses (or any other method) were applied to discriminate between fluxes from one field or the other, or did you just take time windows with no variation in wind direction? You refer to "mean wind direction" (L152-153), but this doesn't necessarily imply the wind came solely from one of the plots. Indeed, there is

an increase in the N2O fluxes in the tilled plot at the same time as the partner, untilled plot was fertilized (first half of June, roughly), so further explanations are needed to exclude potential artifacts wrt that matter. 2) I also miss information on the spatial and temporal coverage using the three techniques. What was the distribution, how many measurements did you do with each method? Probably a summarizing table is the best option, along with more comprehensive information in the Figure captions (e.g. in Fig 4, see below for specific details). Furthermore, do you have simultaneous measurements? Did you appreciate any offset between techniques (it seems so in Fig 5) suggesting you may have a bias depending on which technique you used? Finally, 3) you may need to discuss your gap filling method a bit more: for example, in Figure 6 I see two big mismatches after fertilization, I don't know whether this is due to the subjective adjustment of the degree of smoothing (L206-207) or to some other reason. Did you take observed values or GAM in Fig 7? it is not that clear to me.

Additional minor details:

L137. Remove "See"

L157: Write "N2O emissions " or "N2O emission rates" and not only "N2O"

L185. Maybe I made a mistake when converting units, but I think your detection limits are 0.02 and 0.2 nmol m-2 s-1. 1 nmol N2O m-2 s-1 equals roughly 100 ug N2O-N m-2 h-1.

L212: What is the length of the time steps (30 minutes)?

L218: Strictly speaking, you need to mention Figure 2b as well.

L247. It is stated that the tilled field was not fertilized the previous year. Was the untilled? If there are differences, you should report them in line 95.

L323-334: Don't you have any record on differences of soil water content, bulk density, etc, between tilled and untilled plots? Including these data could feed and improve this section.

L336. Write "Gap filling of N2O flux measurements is challenging" or similar. Gap filling itself is easy (a linear interpolation may be sufficient), but doing it in a sound manner, this is the problem.

L347: This data gap due to instrument failure/unavailability is not to be seen in Figure 4, at least not for two weeks. In L246 you mention a one-week long data drop. Please clarify.

L348: Which type of uncertainty do you refer to? This period of time is showing the smallest uncertainty according to the Figure 7, what makes your statement inconsistent. See also my comment below to Figure 7 on how to show uncertainty

Table 1: Grazing info should be included here, as it is part of the field management.

Fig 2: soil temperature at which depth? In the tilled or the untilled plot? Any soil water content data available?

Is Figure 5 merely a zoom-in into Figure 4? It looks like, but after a closer sight, many static chamber symbols are missing in Fig 5 compared to Fig4. Please clarify and eventually merge the Figures in a single one. Further, please specify what are the values showed in Figs 4 and 5 (averages of # chambers, single chamber values?).

Figure 6. Describe the content of the figure ("mean N2O fluxes as determined by Eddy cov, static and dynamic chambers and estimated N2O flux by using the general additive model from both …" or similar).Are the chamber results the average of # chambers? Please specify this info. Make clearer the difference between the tillage and fertilizer dates by using lines of different color/shape. Again, I have the impression the data shown is "basically" the same as on Fig 4, but with some differences (e.g. there are several dynamic chamber fluxes of 4-8 nmol m-2 s-1 right after tillage in Fig4, or static chamber fluxes after the second fertilization application in 4a, which are gone I Fig 6). By the way, replace "October" by "September" (also in Fig 7).

Figure 7. I guess uncertainty from a time step should propagate to all future dates (in

other words, when you estimate cumulative fluxes, you need to sum up the uncertainty of the precedent values).

---

## Short Comment (SC1) · 8 Mar 2016

This manuscript is a study about the effects of tillage on N2O emissions on Scottish grasslands. It covers an important topic regarding climate change with implications on land use change. However, several points were not clear (described below) and should be explained in more detail in the text. Based on these, I recommend revising the paper.

The study does not specify why 3 different measurements of N2O fluxes were used. Why not only use one method, probably dynamic chamber since it was not constrained by wind patterns (eddy covariance) or by agricultural activities (static chamber in tillage field)? Why was eddy covariance mast a primary method of data collection despite being able to take measurements only from one field at a time especially under the

probability that 85% of wind would come from southwest as you mention (and low overall coverage – 27 and 25%)? Explanation and rationale behind these decisions would be useful.

There is an issue with the estimated emissions of N2O due to tillage (1.08 $\pm$ 0.14 kg N2O-N ha-1). Its calculation was based on the assumption that N2O emissions are constant throughout the year with no tillage or fertilizers. How were other factors that might have influenced baseline N2O emissions throughout the year accounted for in this calculation? A comparison could have been made with untilled field if it was not fertilized throughout the study period along with tilled field. Is there a reason why the land was not divided into 3 fields where the additional one would have been the control field with no tillage and no fertilizers?

There seems to be a correlation between spikes in N2O shortly after fertilization in untilled field and N2O in tilled field. How do we know that the fertilization did not influence also N2O fluxes in tilled field, e.g. by wind which would blow N2O from untilled to tilled field and after that it would be registered by eddy covariance mast as from tilled?

Net effect of tillage could be included in the study by subtracting increased uptake of nitrogen by growing grass post-tillage from increased emissions caused by tillage. This would show an overall impact of tillage.

---

## Author Comment (AC1) · 12 Apr 2016

Dear reviewers,

We would like to thank each of you for your constructive comments which have helped improve the quality of our manuscript. We have made corrections where necessary and hope to justify why we have chosen in some circumstances not to edit the manuscript in this response.

Since publishing the original manuscript it has come to our attention that there was a small offset in measurement time between the flux measurements and meteorological measurements that were used in the GAM gap filling process (several hours). This error has been corrected for in the final response, and as such, the majority of the figures and reported values in the text have been altered. The changes do slightly alter the results of the experiment; however, the overall outcome is very much the same.

As many of the reviewers' comments were similar in nature I have compiled all of the replies into one document in which I can discuss all concerns.

Several of the reviewer's comment on wind direction and the source of the fluxes, so we carried out a more in depth footprint analysis of the eddy covariance fluxes. We found that a handful of the measurements did not cover a large enough area for an effective estimate of flux from the fields. This was partly the result of the friction velocity cut-off being too low in our QC. We increased this from 0.05 to 0.1 m s-1 to help correct for this issue. Any other footprint that we did not believe was adequate was removed.

There was primarily a concern that the high fluxes measured from the tilled field (south) may have been partly due to advection from the fertiliser event on the adjacent field. We present the footprints of some of the highest fluxes measured in this period below as examples.

Having examined the footprint in greater detail we are confident that the fluxes reported for each of the fields are valid. We do not feel that adding the footprint plots to the manuscript is necessary as there are hundreds of individual footprints and an averaged plot tells us nothing in particular about the validity of individual measurements. (note that the google maps picture above is years out of date, the area to the right hand side of the site was no longer under work during measurements). We have added that a footprint analysis has been carried out to the methodology section of the manuscript.

L153 "A footprint analysis was carried out in which we visually checked individual footprint plots of each 30 min flux. Any flux measurements that were very small or overlapped the different fields were removed from the dataset"

Another comment echoed by the reviewers was that we should try to estimate annual emissions from the fields. During the study we measured from the site for as long as was possible. The QCL instrument was fairly new and had been booked for multiple experiments. The focus of this particular experiment was supposed to be on the short term effect of tillage (several months). In numerous previous experiments carried out at the field site it is rare to see any significant N2O flux activity with the exception of fertiliser application. As flux measurements had returned to near zero/background flux magnitude by September we assume that any immediate effect of the tillage event has taken course.

With any published N2O flux event there is always calls for annual emission factors, but when annual fluxes are estimated from limited datasets there are often concerns about the validity of such estimates due to the unpredictable nature of N2O fluxes. With the data available to us we cannot please everyone. The data presented in this paper is the full data set we have available and we are faced with the option of either making annual cumulative flux estimates based on a combination of the data and loosely based assumptions, or shorter estimates based on the actual measured data. In the manuscript we focus on the measured data. As with all N2O studies, uncertainties in any gap filling beyond actual measurements are likely to be large and any annual flux estimate would be largely theoretical in nature.

**Comments to Reviewer 1**

Major Comments: (1) There are yet not too many N2O flux datasets from grassland available and even less from tillage. I am wondering why the authors make such a short story out of this valuable dataset. This study is based on three different techniques and the actual discussion on the capability of these techniques to capture the same results differ. Even though the temporal pattern seems to be captured well, there seems also a stable offset (Figure 5) between EC and chamber measurements. This should be further elaborated.

The three different methods were used to gather as much measurement data as was possible with the limited resources at our disposal during the experiment. We have added a small bit of text:

L129 "The mixture of methods were used to try to obtain as many measurements as practically possible both temporally and spatially during the experiment."

It is very likely that the measurements from the different methodologies could have potential offsets or bias when compared with each other, but explaining this would take a significant number of measurements, more than we have for this particular study. This is currently the topic of a further paper that we are in the process of writing at the moment in which spatial and temporal gap filling methods and uncertainties are being investigated at multiple sites for different flux measurement methods using a Bayesian statistics approach. The content is beyond the scope of this paper on tillage.

**(2) When using an approach as in this study, I wonder how the authors assure that the flux from the specific 30min of EC observations originated for consecutive 30min from the area of observation? Did the authors include specific tests to proof that the approach is suitable?**

Referring back to the footprint plots (above) we have checked to see if the footprint covers a valid area of the field being measured. The wind direction at the site comes from predominantly one field or the other. Anything in the middle was stripped out of the data set, although the majority of this was caught first time round. Most corrections have been made regarding very small footprints rather than field overlap.

**(3) A gapfilling approach for N2O data is presented which is suitable to be applied at many other N2O EC sites. Therefore I suggest to further elaborate on this.**

It is difficult to elaborate on the method any further than to explain in detail the workings of the mcgv package for R which is referenced in the text. We simply used the package as it was intended in a way which allows us to gap fill and estimate uncertainties. More information can be found at:

https://cran.r-project.org/web/packages/mgcv/mgcv.pdf

(4) The authors provide cumulative N2O fluxes for 175days of gapfilled N2O data. However it remains unclear how these values shall be set in context with other studies where growing season budgets or annual budgets are presented. Is there any additional data or knowledge that could lead to annual numbers?

See previous description of annual flux estimates.

**L58ff: the number provided are focusing on a specific time period but this is not indicated, please clarify. Also write kg instead of Kg**

Added "annual" to text and changed Kg to kg

L62: replace ";" by "."

Done.

**L63: aeration instead of aerobicity**

Done

L124: aboveground

Done

**L131: I suggest to add manual or automatic since this is unclear**

"Manual" added

**L134: is that an experiment or rather a survey? see also Eugster and Merbold et al. 2015, SOIL 1 for clarification**

We have tilled a field and left one un-tilled specifically as an experiment to compare the two so I believe that this is more of an experiment rather than a survey, although I could be wrong.

**L134ff: ">30" - dynamic doesn't mean per se automatic, but it remains unclear how you achieved this, so the needed information should be provided here already.**

Added "manual" but the method is highly detailed in the reference included in the methodology

Cowan, N. J., Famulari, D., Levy, P. E., Anderson, M., Bell, M. J., Rees, R. M., Reay, D. S. and Skiba, U. M.: An improved method for measuring soil N2O fluxes using a quantum cascade laser with a dynamic chamber, Eur. J. Soil Sci., 65(5), 643–652, doi:10.1111/ejss.12168, 2014a.

**L152: did you check whether the wind was blowing from the specific area for the specific half-hour?**

See footprint analysis above

**L155: Specs should be given for rainfall, temperature etc measurements: where, how etc? or refer to another paper (Drewer et al. Plant and Soil for instance**

**Text changed to**

L155 "Standard meteorological variables (rainfall, air temperature and soil temperature) were recorded by a tipping bucket, thermometers (2 m height & 10 cm depth) and TDR soil moisture probe at 10 cm depth. These measurements were made adjacent to the flux tower at the site."

**L176: ID?**

Changed to inner diameter

**L177: "prior to measurement" - how long before?**

Added text

"at least 15 mins"

**L189: how can you assure that this method is suitable to fill the gap? Details, validation is needed.**

Ultimately we can't be sure the method is suitable. There is no absolute way to assess this. The same applies to all  $N_2O$  gap filling methods. The advantage of the GAM method over other methods is that it provides uncertainty in the flux estimate constrained statistically by the meteorological measurements, a weakness that linear interpolation or smoothing cannot provide.

**L197: I disagree – Spacsys and PaSIM don't perform too badly – So I suggest to give a reference for your statement**

I disagree. Although these types of models may be improving, they are still very limited. Often these models are so complex that they can be adjusted to suit any data set measured from a specific site which gives the appearance of a successful prediction. Published papers on these models tend to highlight any good correlations and ignore the inconsistencies. Even the Spacsys and PaSIM models are relatively weak when it comes to accurately predicting flux behaviour.

**L199ff: Can extend this section so that others can choose a similar approach?**

I agree it is difficult method to understand from the text in the manuscript, but to go into details would be no more than an explanation of the mgcv package itself. We only use an already available tool in the manuscript. bThe information required to use the R package is found at the CRAN website mentioned before. The reference in the paper is also useful:

Wood, S. N.: Generalized additive models: an introduction with R, Chapman & Hall/CRC, Boca Raton, FL., 2006.

**L202: same model terms compared to what?**

Changed text to "same environmental parameters".

**L232: Where is this shown, similar on which time interval? stats?**

A description of the similarities is written directly after the statement in the text.

"In both cases, around 90 % of fluxes were below 0.5 nmol  $m^{-2} s^{-1}$  (Figure 4)."

**L234: detection limit of your system?**

We assume the detection limit of the system is similar to the magnitude of the noise in the measured fluxes when flux is near zero. However, isolating a single value from the real variations in low flux conditions is not entirely possible and it is likely that the flux detection limits vary depending on wind conditions. We estimate that the detection limit of the system is between 0.25 and 1 nmol m-2 s-1 but this is purely subjective.

**L236: what is your background flux and how was it determined?**

Background flux is the average of the fluxes measured before any management was carried out. Generally speaking it is any value below 0.5 as 90 % of the measurements fell below this value as described in the text.

"Before the tillage event, N2O fluxes were similar in the tilled and untilled fields. In both cases, around 90 % of fluxes were below 0.5 nmol  $m^{-2}$  s-1 (Figure 4)."

**L237ff: how comparable are these cumulative values to other studies? is that the main growing season, how will the fluxes be during the remaining season?**

Cumulative flux studies on grasslands in the area typically measure  $N_2O$  fluxes close to zero for background measurements and approximately 0 to 5 % as emission factors for fertiliser events. Our estimates are similar in this respect, but a direct comparison of such a wide range of results which have been reported in the past limits the importance of these comparisons. We compare with the IPCC value of 1 % in the discussion section.

**L249: were sheep in the untilled field?**

Occasionally, although their presence seems negligible in the flux measurements.

Added information to Table 1

**L275: "were estimated" or "estimated by least square method?"**

Fixed

**L279: what do you try to state with the estimated?**

The short term contribution to N2O fluxes as a direct effect of the tillage event.

**L283: Was all plant material incorporated or was the field grazed beforehand? when were the biomass samples taken?**

The samples were taken before the tillage event. The grass had been grazed beforehand. Added text to clarify:

"made prior to tillage"

**L287- L295: This is basically a repetition of already presented results.**

Agreed, but it helps the reader remember without having to look back through the manuscript.

**L311: What is your definition of large in comparison to 0.5nmol m-2 s-1**

Added text

L318 "Duplicate fertiliser applications to the fields in autumn result in a difference in cumulative fluxes of approximately 0.48 kg  $N_2$ O-N ha-1."

**L321: What about N deposition and mineralisation the soil?**

This would be assumed to be the same for the adjacent similarly managed fields and not a significant enough source to effect the flux measurements in a way that would bias the comparison of them.

**L325: any proof or measurement here? Pictures, CO2 fluxes or likewise?**

It was clear to see just by looking at the fields; however, we have no photographs to show.

**L340: include Butterbach-Bahl et al. 2013 - review here**

Added

**L352: please explain? references?**

Added Eugster and Merbold, 2015

**Figure 1: Was there any bias to the dynamic chamber if these are soo close to the cabin?**

Probably, but there was no alternative. There is no more bias in this method than that of autochamber approaches set up to do a similar task. Similarly the placement of the eddy covariance tower could be argued as bias to a particular area of the field.

**Figure 2: height/depth of measurements?**

Text added

**Figure 3: so everything adds up to 100%, correct? NNE high winds only in 2% of the data? Correct?**

I don't really understand the question. Yes everything adds up to 100 %. The wind in the area is very limited to SW and NE due to the topography of the site. Wind rarely blows from any other direction (as can be seen in the plot).

**Figure 4: why are the EC fluxes all in one line after the fertilization event, are these 30min data or which averaging interval did you use? triangles and squares in grey are difficult to distinguish. What about adding an axis break including the higher fluxes? how much higher than**

The time break between the fluxes is 30 mins. Because this is relatively short compared to the full range of the axis it appears they are all in the same x position. It's just the resolution of the plot, not much can be done without stretching the plot out very long.

Text added:

L242"Four high individual chamber measurements were measured in the days immediately after the second fertilisation event in the un-tilled field. These measurements (13.3, 19.5, 34.8 and 50 nmol  $m^{-2} s^{-1}$ ) are not included in Figures 4 or 5 in order to keep the scale manageable."

**Figure 5: I suggest in all figures to change the unit indication to "()" instead of "/"**

I will leave aesthetic choices to the editor. In a previous submission to the journal I was told the opposite, but I am happy to make changes if necessary.

**Figure 6: This looks really good but I cannot see the GAM line in much detail I hope the comments are helpful and look forward to seeing this paper being published soon.**

Changed the shading/colour scheme of the plot to try to improve how the GAM prediction fits with the other data points. The points on the plot are just a duplicate of Figure 4 with the GAM fit on top.

**Comments to Reviewer 2**

Authors should also try to reflect how they could come up with annual fluxes and what would be the uncertainty there (e.g. are there soil freeze-thaw events?); in other words, authors should mention what are we (likely) missing in the 190 days which were not measured.

As described above, we do not have these measurements and therefore annual flux estimates could only be based loosely on what we would expect to be normal for the site. We tend not to see any effect of freeze/thaw events at the site as snow tends only to last several days at a maximum during winter. Typically fluxes measured in past experiments remain low, close to the detection limits of the static chamber measurements. No measurements were carried out for the remainder of the year in which tillage took place.

**Spatial and temporal variability is somehow addressed in the discussion (L350-357), but could be more elaborated, and some information on spatial variability (as assessed by chamber measurements) should be included.**

We agree that spatial variability in chamber measurements is a large source of error in  $N_2O$  flux measurements; however, addressing the issue goes well beyond the scope of this manuscript. Further work on the issue is underway (see answer to reviewer 1, comment 1). We add the chamber measurements to our GAM gap filling model which works empirically to establish a fit in the data. Theoretically although spatial variability is a large source of error, it is also random in nature and should not bias the results of the cumulative flux estimates presented in this manuscript.

I recommend making a stronger point in the discussion addressing the implications of tillage for accounting of N2O fluxes in the context of grassland management, including the fundamental fact that tillage (responsible for an additional release of 1.08 kg N ha-1) "substitutes" one fertilization event (responsible for 0.72 kg N ha-1, L317). For instance, confidence intervals of N2O emissions from tilled and untilled plots overlap by the end of the season (Figure 7), so I presume differences in cumulative fluxes were not statistically significant; that should be also addressed clearly in the work.

Although the confidence intervals overlap it is difficult to say that the tillage "substituted" a fertilisation event. If uncertainty analysis was carried out properly on all  $N_2O$  studies it is likely that most of them would never report anything statistically significant. The nature of  $N_2O$  gap-filling prevents us from knowing the true uncertainty in our measurements and so it is likely even after all the effort we put in, these error bars are probably smaller than they should be. The paper clearly states that the effects of grassland tillage can be large in some cases, although we still don't really understand why. As most tillage events have been measured using chamber methods in the past it is entirely possible that the effect of tillage we observed in this experiment may also occur on arable fields and has just not been well documented in previous experiments.

**Added text**

L359: "a value akin to a 105 Kg-N fertiliser application according to IPCC emission factor estimates"

My second major point is oriented to methods. I missed some necessary explanations on 1) whether footprint analyses (or any other method) were applied to discriminate between fluxes from one field or the other, or did you just take time windows with no variation in wind direction? You refer

to "mean wind direction" (L152-153), but this doesn t necessarily imply the wind came solely from one of the plots. Indeed, there is an increase in the N2O fluxes in the tilled plot at the same time as the partner, untilled plot was fertilized (first half of June, roughly), so further explanations are needed to exclude potential artifacts wrt that matter.

See footprint plots above.

2) I also miss information on the spatial and temporal coverage using the three techniques. What was the distribution, how many measurements did you do with each method? Probably a summarizing table is the best option, along with more comprehensive information in the Figure captions (e.g. in Fig 4, see below for specific details). Furthermore, do you have simultaneous measurements? Did you appreciate any offset between techniques (it seems so in Fig 5) suggesting you may have a bias depending on which technique you used?

Regarding comparing methods, see our response to reviewer 1, comment 1. The chamber measurements were carried out intermittently when staff and weather allowed. The dynamic chamber measurements were not always in the same place and different numbers were recorded on various dates. The chambers are simply used as an additional source of measurement data to help gap fill the eddy covariance measurements as much as possible. Any spatial analysis of N2O at that scale from such few measurements is highly limited to the point that the analysis is fairly meaningless. As mentioned above, there is likely an offset between the methods, but there are too few measurements to assess the true relationship between them. This matter is currently a work in progress at a much larger scale with many data sets and field sites.

Finally, 3) you may need to discuss your gap filling method a bit more: for example, in Figure 6 I see two big mismatches after fertilization, I don t know whether this is due to the subjective adjustment of the degree of smoothing (L206-207) or to some other reason. Did you take observed values or GAM in Fig 7? it is not that clear to me.

The GAM is empirical in nature. We just use it to fit the predictions as well as possible to the data. Any mismatches should be unbiased in nature as long as we keep the gap filling constrained between actual measurement points (i.e. if it's too high somewhere, it will likely be too low somewhere else). The plots will look different now as the data has gone through further QC controls and the met data has been realigned temporally.

See the reference for more details on how the GAM functions.

Wood, S. N.: Generalized additive models: an introduction with R, Chapman & Hall/CRC, Boca Raton, FL., 2006.

**Additional minor details:**

L137. Remove "See"

Done

L157: Write "N2O emissions " or "N2O emission rates" and not only "N2O"

Changed to "N2O fluxes"

L185. Maybe I made a mistake when converting units, but I think your detection limits are 0.02 and 0.2 nmol m-2 s-1. 1 nmol N2O m-2 s-1 equals roughly 100 ug N2O-N m-2 h-1.

Static chambers can have a fairly poor detection limits due to noisy GC's. The values are from the papers mentioned in the text (Cowan et al., 2014a, 2014b).

**L212: What is the length of the time steps (30 minutes)?**

Added text "30 min interval"

**L218: Strictly speaking, you need to mention Figure 2b as well.**

Added text "The annual variation in temperature was fairly typical of the field site (Figure 2b)"

**L247. It is stated that the tilled field was not fertilized the previous year. Was the untilled? If there are differences, you should report them in line 95.**

The text says the field was not fertilised 'since' the last year. Both fields were fertilised the previous year at the same time.

L323-334: Don t you have any record on differences of soil water content, bulk density, etc, between tilled and untilled plots? Including these data could feed and improve this section.

Unfortunately nothing of a publishable standard.

L336. Write "Gap filling of N2O flux measurements is challenging" or similar. Gap filling itself is easy (a linear interpolation may be sufficient), but doing it in a sound manner, this is the problem.

**Text changed**

L347: This data gap due to instrument failure/unavailability is not to be seen in Figure 4, at least not for two weeks. In L246 you mention a one-week long data drop. Please clarify.

The QCL was removed for a week (5 days) and there is a gap in the data, although it may be difficult to spot. It is directly after the first fertiliser peak in the North field.

Changed text to "five days" instead of "week".

**L348: Which type of uncertainty do you refer to? This period of time is showing the smallest uncertainty according to the Figure 7, what makes your statement inconsistent. See also my comment below to Figure 7 on how to show uncertainty**

Missing data contributes to uncertainty in a way that is incalculable. We can only fit statistical models to the data we have, when temporal variability is high there is very little we can do to estimate it, or the uncertainty in our estimates. There is likely a greater uncertainty in the estimates than we have calculated, but it is impossible to quantify with the data we have.

**Added text**

L345 "which is not possible to estimate"

**Table 1: Grazing info should be included here, as it is part of the field management.**

Done

**Fig 2: soil temperature at which depth? In the tilled or the untilled plot? Any soil water content data available?**

L153 "Standard meteorological variables (rainfall, air temperature and soil temperature) were recorded by a tipping bucket, thermometers (2 m height & 10 cm depth) and TDR soil moisture probe at 10 cm depth. These measurements were made adjacent to the flux tower at the site."

Is Figure 5 merely a zoom-in into Figure 4? It looks like, but after a closer sight, many static chamber symbols are missing in Fig 5 compared to Fig4. Please clarify and eventually merge the Figures in a single one.

Further, please specify what are the values showed in Figs 4 and 5 (averages of # chambers, single chamber values?).

Figure 6. Describe the content of the figure ("mean N2O fluxes as determined by Eddy cov, static and dynamic chambers and estimated N2O flux by using the general additive model from both . . ." or similar). Are the chamber results the average of # chambers? Please specify this info. Make clearer the difference between the tillage and fertilizer dates by using lines of different color/shape. Again, I have the impression the data shown is "basically" the same as on Fig 4, but with some differences (e.g. there are several dynamic chamber fluxes of 4-8 nmol m-2 s-1 right after tillage in Fig4, or static chamber fluxes after the second fertilization application in 4a, which are gone I Fig 6).

To avoid any confusion I have removed Figure 5. By enlarging the other figures it is clear to see how the measurements change after the events. Originally the chamber fluxes had been averaged into 30 minute segments but I believe the plots look better now and are easier to understand.

**Text added**

L242 "Four high individual chamber measurements were measured in the days immediately after the second fertilisation event in the un-tilled field. These measurements (13.3, 19.5, 34.8 and 50 nmol  $m^{-2} s^{-1}$ ) are not included in Figures 4 or 5 in order to keep the scale manageable."

**By the way, replace "October" by "September" (also in Fig 7). Figure 7.**

Done

**Reviewer 3**

**The study does not specify why 3 different measurements of N2O fluxes were used. Why not only use one method, probably dynamic chamber since it was not constrained by wind patterns (eddy covariance) or by agricultural activities (static chamber in tillage field)?**

The chamber methods were performed manually and could not provide the consistent temporal coverage that the eddy covariance provided. Had the eddy covariance measurements not taken place we would have missed many of the peaks in  $N_2O$  fluxes.

**Why was eddy covariance mast a primary method of data collection despite being able to take measurements only from one field at a time especially under the probability that 85% of wind would come from southwest as you mention (and low overall coverage – 27 and 25%)?**

Originally we planned to only measure the tilled field; however the wind was unpredictable that year and came from NE more than normal. Eddy covariance gives 30 min measurements constantly providing better temporal and spatial coverage than chambers can making it a more suitable method to use in such an experiment. The down side is that after quality controls, such as data when wind speeds are too low for the basic eddy covariance assumptions to make sense are removed, we lose a lot of data. Typically about a 30 % data coverage after QC is not rare.

There is an issue with the estimated emissions of N2O due to tillage  $(1.08 \pm 0.14 \text{ kg N2O-N ha-1})$ . Its calculation was based on the assumption that N2O emissions are constant throughout the year with no tillage or fertilizers. How were other factors that might have influenced baseline N2O emissions throughout the year accounted for in this calculation? A comparison could have been made with untilled field if it was not fertilized throughout the study period along with tilled field.

**Is there a reason why the land was not divided into 3 fields where the additional one would have been the control field with no tillage and no fertilizers?**

Although the field was tilled as part of the experiment, it was agreed with the farmer as the field required it anyway. The fields are run by a real farm and as such we cannot interfere too much with them. The wind direction comes primarily from two directions so we could only really have two treatments. The un-tilled field required fertiliser to provide the grass for the farm as we could not ask the farmer to lose two grazing pastures for us in one year. The baseline was simply an assumption that fluxes would remain similar to what they were before the tillage event (i.e. the average).

There seems to be a correlation between spikes in N2O shortly after fertilization in untilled field and N2O in tilled field. How do we know that the fertilization did not influence also N2O fluxes in tilled field, e.g. by wind which would blow N2O from untilled to tilled field and after that it would be registered by eddy covariance mast as from tilled?

See footprint analysis above

Net effect of tillage could be included in the study by subtracting increased uptake of nitrogen by growing grass post-tillage from increased emissions caused by tillage. This would show an overall impact of tillage.

It could, but we don't have the grass measurements or N uptake.

Hopefully our response to these comments is adequate for the reviewers. We are happy to answer any further questions in the future regarding our work.

Sincerely,

Dr Nicholas Cowan

---

## Author Response (AR1)

Dear Editor/Reviewers,

Having had the time to revise the manuscript "The influence of tillage on $N_2O$ fluxes from an intensively managed grazed grassland in Scotland", we have improved the text based on the reviewer's comments. An improved approach to the data analysis and better description of the footprint analysis has altered several of the results/plots in the manuscript and has required a significant amount of re-writing.

**Data Analysis**

A correction was applied to the Gill Windmaster sonic measurements. The manufacturers recently announced that the sonics require a correction to the vertical wind (w) component. This correction was applied to our raw data and fluxes were re-processed.

Concern that the fetch of the eddy covariance measurements could not distinguish between the separate fields was echoed between multiple reviewers. We hope that in our improved explanation of the footprint analysis that we have addressed these concerns. Having re-processed the data we have now included a figure showing the difference in $CO_2$ fluxes between the fields after tillage. The clear difference between these measurements from the fields shows that our footprint method is adequate.

The u* filtering of the eddy covariance fluxes was further investigated and no relationship could be found between u* and flux. As there was no significant relationship we could not justify any particular cut-off point at which to filter. As an alternative we looked at the footprint plots. Any footprints that were very small, and not providing an adequate fetch in either field were removed. This retained more measurement data points than using the basic u* filtering which subsequently improved gap filling methods, primarily in the tilled field.

Reviewers had some concern about confusion between averaged flux values and individual flux measurements in the figures. To address this we have removed figures with averaged fluxes. Every individual measurement is now treated equally in the GAMM model and all measurements are shown in the plots.

**Results**

As the calculated fluxes and the fitted gap-filling model is now different, the reported results have also changed. The individual measured fluxes changed relatively little, but the gap-filling and cumulative flux estimates are significantly different compared to our initial submission.

The main difference in the results is that the emissions associated with the fertilisation events in august are almost identical in magnitude (0.76 and 0.77 kg-N ha$^{-1}$). Our previous discussion about the difference in fluxes between the fields has subsequently been removed.

The paper is now much more focussed on the emissions from the tilled field in the months after tillage. The study highlights a potentially large seasonal hotspot of $N_2O$ emissions from agricultural sources which should be investigated and included in regional/national inventories.

I hope that these edits have improved the manuscript and look forward to your input.

Sincerely

Nicholas Cowan

[revised manuscript text omitted]

---

## Author Response (AR2)

Dear editor,

We would again like to thank you and the reviewers for your input in the editing process of this manuscript. We agree with the majority of the further suggestions and have altered the manuscript to accommodate them.

The reviewer is correct in spotting that fluxes occur before the fertilisation plotted on the figure. This was due to the date of the fertilisation in the plot being wrong rather than early fluxes. This has been corrected for.

With regards to the issue with the reported detection limits, there is a bit of uncertainty over what the "detection limit" actually represents. The reviewer is correct in stating that the uncertainty in individual measurements reported in Cowan et al 2014 are 2 and 20 $\mu gN_2O$-N $m^{-2}$ $h^{-1}$, which roughly equals 0.02 and 0.2 nmol $m^{-2}$ $s^{-1}$, respectively. Investigating the detection limits further in Cowan et al 2014b we double this value (i.e. double the standard deviation of a zero flux measurement.) This is why the reported detection limits are larger than in the initial paper.

With regards to editing the figures, it is very difficult to contrast the black and grey colours in the plots without keeping them as they are. In Figure 5 the black dots stand out more and it is clear which of the measurements are eddy covariance and which are chambers. In Figure 6, the contrast between the GAM prediction and the measurements is more important. If the editor requests it, then we can try again to make the plots clearer with consistent colours/shapes; however for a relatively minor change, we hope this is not an issue.

I believe all other points in the reviewers replies have been edited and corrected for.

Sincerely,

Nick Cowan

[revised manuscript text omitted]